# Circulating myomiRs in Muscle Denervation: From Surgical to ALS Pathological Condition

**DOI:** 10.3390/cells10082043

**Published:** 2021-08-10

**Authors:** Irene Casola, Bianca Maria Scicchitano, Elisa Lepore, Silvia Mandillo, Elisabetta Golini, Carmine Nicoletti, Laura Barberi, Gabriella Dobrowolny, Antonio Musarò

**Affiliations:** 1Laboratory Affiliated to Istituto Pasteur Italia—Fondazione Cenci Bolognetti, DAHFMO-Unit of Histology and Medical Embryology, Sapienza University of Rome, Via A. Scarpa, 14, 00161 Rome, Italy; irene.casola@uniroma1.it (I.C.); elisa.lepore@uniroma1.it (E.L.); carmine.nicoletti@uniroma1.it (C.N.); laura.barberi@uniroma1.it (L.B.); 2Sezione di Istologia ed Embriologia, Dipartimento di Scienze della Vita e Sanità Pubblica, Fondazione Policlinico Universitario A. Gemelli IRCCS, 00168 Roma, Italy; Biancamaria.Scicchitano@unicatt.it; 3Institute of Biochemistry and Cell Biology, National Research Council, Monterotondo scalo, 00015 Rome, Italy; silvia.mandillo@cnr.it (S.M.); elisabetta.golini@cnr.it (E.G.); 4Scuola Superiore di Studi Avanzati Sapienza (SSAS), Sapienza University of Rome, 00185 Rome, Italy

**Keywords:** circulating miRNA, ALS, muscle denervation

## Abstract

ALS is a fatal neurodegenerative disease that is associated with muscle atrophy, motoneuron degeneration and denervation. Different mechanisms have been proposed to explain the pathogenesis of the disease; in this context, microRNAs have been described as biomarkers and potential pathogenetic factors for ALS. MyomiRs are microRNAs produced by skeletal muscle, and they play an important role in tissue homeostasis; moreover, they can be released in blood circulation in pathological conditions, including ALS. However, the functional role of myomiRs in muscle denervation has not yet been fully clarified. In this study, we analyze the levels of two myomiRs, namely miR-206 and miR-133a, in skeletal muscle and blood samples of denervated mice, and we demonstrate that surgical denervation reduces the expression of both miR-206 and miR-133a, while miR-206 but not miR-133a is upregulated during the re-innervation process. Furthermore, we quantify the levels of miR-206 and miR-133a in serum samples of two ALS mouse models, characterized by different disease velocities, and we demonstrate a different modulation of circulating myomiRs during ALS disease, according to the velocity of disease progression. Moreover, taking into account surgical and pathological denervation, we describe a different response to increasing amounts of circulating miR-206, suggesting a hormetic effect of miR-206 in relation to changes in neuromuscular communication.

## 1. Introduction

Amyotrophic Lateral Sclerosis (ALS) is a fatal neurodegenerative disease characterized by motoneuron degeneration, muscle atrophy and denervation [1]. A total of 20% of familial cases of ALS exhibit mutation in the gene coding for the Superoxide Dismutase 1 (SOD1).

One of the most characterized animal models, which displays several pathological signs of ALS, is the transgenic mouse, which ubiquitously expresses the mutated form of the SOD1 human gene. In particular, two transgenic SOD1^G93A^ mouse models have been generated to develop different velocities of disease progression: the SOD1^G93A^ high-copy transgenic mice, which exhibit 20–25 copies of the human muted gene, resulting in a rapid pathological decline, and the SOD1^G93A^low-copy mice, exhibiting 8–10 copies of the human muted gene and characterized by a slower pathological decline.

The SOD1^G93A^ high-copy transgenic mice display the first symptoms of the disease at 90 days of age and complete paralysis, followed by death at 120 days of age [2]. The SOD1^G93A^ low-copy mice show the first symptoms of the disease at 150 days of age, and at 220 days exhibit loss of muscle mass and strength and progressive paralysis, which leads to death at 250 days of age [3].

Several molecular mechanisms have been proposed to account for the pathogenesis of the disease; among these, microRNAs represents both disease biomarkers and potential pathogenetic factors that can either exacerbate or delay the disease. MicroRNAs are evolutionary conserved non-coding RNAs, with a post-transcriptional regulatory role. Myomirs, such as miR-206 and miR-133a, are microRNAs produced in skeletal muscle, and play a key role in tissue homeostasis. They are involved in muscle proliferation, differentiation and regeneration processes [4], and are differently modulated in skeletal muscle denervation and re-innervation; indeed, studies in rat models have demonstrated that surgical denervation, followed by re-innervation, increases miR-206 and miR-133a muscle levels, and these remain high until one month after surgery [5]. 

The role of miR-206 in ALS has been deeply studied; in particular, it has been demonstrated that muscle expression of mir-206 can delay ALS progression and promotes regeneration of neuromuscular synapses in ALS mouse models and patients [6,7].

Recent studies have demonstrated that tissue-specific microRNAs can be released in blood circulation and that their expression is significantly deregulated in several human disorders, including ALS, suggesting the possibility to use circulating microRNAs as diagnostic and prognostic biomarkers for degenerative diseases [8,9,10]. In a recent work, we have demonstrated that the serum levels of miR-206 are significantly modulated during ALS pathology, whereas circulating miR-133a remain low throughout the course of the disease. Further, we demonstrated that high levels of circulating miR-206 and miR-133a can predict a slower clinical decline of ALS patients, indicating that miR-206 and miR-133a represent potential prognostic markers for ALS pathology [11].

The functional role of miR-206 and miR-133a modulation in muscle tissue or in blood circulation during muscle denervation and re-innervation is still unknown. Therefore, in the present work we aimed to evaluate the expression levels of both miRNAs in murine skeletal muscle and serum samples of transgenic (tg) SOD1^G93A^ and WT mice after surgical denervation, associated or not with muscle re-innervation. We demonstrated that surgical denervation results in the decreased expression of miR-206 and miR-133a, while miR-206 but not miR-133a was upregulated during the re-innervation process.

Further, to investigate the role of miR-206 and miR-133a in ALS disease, we quantified by absolute RT-PCR the levels of miR-206 and miR-133a in serum samples of high-copy and low-copy SOD1^G93A^ mice. We demonstrated a different modulation of circulating miR-206 and miR-133a during ALS disease, according to the velocity of disease progression. Moreover, the comparative analysis of miR-206 and miR-133a levels in the ALS mouse models and surgical denervated mice demonstrated a different response to increasing amounts of miR-206, suggesting a hormetic effect of circulating miR-206 in relation to changes in neuromuscular communication.

## 2. Materials and Methods

### 2.1. Mice

The experiments were performed on transgenic SOD1^G93A^ high-copy and low-copy mice, provided by Jackson Laboratory, and on C57BL6/J mice. The animals were housed in a temperature-controlled room (22 °C), with a 12:12 h light–dark cycle. Blood and muscles tissues were collected in accordance with the guidelines of the institutional animal facility. SOD1^G93A^ high-copy mice were sacrificed at the onset (90 days) and at the end stage of the pathology (150 days). SOD1^G93A^ low-copy mice were sacrificed at the onset (150 days), at overt signs (220 days) and at the end stage of the disease (250 days). Clinical and behavioral signs were confirmed according to the observations detailed in previous work [2,3].

### 2.2. Animal Surgery

C57BL6/J 5 months old mice were randomly assigned to the control, denervation or reinnervation experimental groups. Mice were anesthetized by intraperitoneal injection of tribromoethanol (0.2 mL/g), and the sciatic nerve was exposed at the mid-thigh level through dorsolateral skin incision. For the denervation group, surgical removal of a 1 cm segment of the proximal nerve was performed, and the stump of the proximal nerve ending was ligated in order to prevent nerve regeneration. For the reinnervation group, surgical removal of the proximal nerve was not followed by nerve ending ligation, in order to allow nerve regeneration. The animals were sacrificed 2, 10, and 30 days after surgery.

### 2.3. Serum microRNAs Analysis

Circulating microRNAs were extracted from 100 µL of serum by miRNeasy Micro Kit (QIAGEN, Hilden, Germany, Cat#1071023). During the extraction, 3.5 µL of Spike-In (Ambion, Thermo Fisher Scientific, Waltham, MA, USA, cel-miR-39-, Cat#4464066, 1.6 × 10^8^ copies/µL) was added to each sample to allow the absolute quantification of microRNAs. Subsequently, 10 ng of the isolated microRNAs were retrotranscribed by TaqMan MicroRNA Reverse Transcription Kit (Thermo Fisher Scientific, Waltham, MA, USA, Cat#4366596), and cDNA was diluted 1:3.3. Finally, 5 µL of cDNA were used for quantitative Real Time-PCR through ViiATM 7 RealTime-PCR System (Applied Biosystems, Thermo Fisher Scientific, Waltham, MA, USA), using TaqMan Universal Master Mix II (Thermo Fisher Scientific, Waltham, MA, USA, Cat#4440040) and specific probes.

To obtain an absolute quantification of microRNAs, for each PCR assay we generated a standard curve of the Spike-In template. For all the samples, the mean value of the Ct of each target microRNA, including the Spike-In, was calculated, and the absolute number of microRNA copies was obtained by comparison with the Spike-In standard curve. Furthermore, we used a mathematical algorithm to take into account the efficiency of the extraction, retro-transcription and amplification procedures to overcome experimental bias and to calculate the real copies number of circulating microRNAs in the sample analyzed.

### 2.4. Muscle microRNAs Analysis

Total RNA was isolated from gastrocnemius muscle by QIAzoLLysis Reagent (QIAGEN, Hilden, Germany, Cat#79306). The yield, quality and integrity of RNA were determined using NanoDrop ND-2000 (Thermo Fisher Scientific, Waltham, MA, USA). Subsequently, 10 ng of RNA were retrotranscribed using TaqMan MicroRNA Reverse Transcription Kit (Thermo Fisher Scientific, Waltham, MA, USA, Cat#4366596), and cDNA was diluted 1:3.3. Finally, 5 µL of cDNA were used for quantitative Real Time-PCR through ViiATM 7 RealTime-PCR System (Applied Biosystems, Thermo Fisher Scientific, Waltham, MA, USA), using TaqMan Universal Master Mix II (Thermo Fisher Scientific, Waltham, MA, USA, Cat#4440040) and specific probeS. Quantitative Real Time-PCR values were normalized for the expression of U6 snRNA, and relative expression was calculated through the 2^−∆∆Ct^ method [12] and reported as fold change.

### 2.5. Statistical Analysis

Statistical parameters are reported in figure legends. Unless otherwise indicated, statistical significance was calculated through an unpaired t-test (* *p*  <  0.05, ** *p*  <  0.005, *** *p*  <  0.0005), and data are represented as mean  ±  SEM. Statistical analysis and ROC curves were performed using GraphPad PRISM 6 software (version 6, San Diego, CA, USA).

## 3. Results

### 3.1. MiR-206 and miR-133a Levels Are Differently Modulated in Serum and Skeletal Muscle during Denervation and Re-Innervation Processes

Mir-206 and mir-133a are involved in several neurodegenerative processes, including muscle denervation. We first evaluated in wild type mice (WT) whether myomiRs’ expression in muscle tissue can be modulated by axon sprouting after denervation. To this purpose, we performed a RT-PCR analysis in the gastrocnemius muscles of wild type mice after sciatic nerve resection, followed by either ligation of the axon terminal, to prevent axon regeneration, or followed by spontaneous collateral sprouting of the sciatic nerve to induce muscle re-innervation.

As described in Figure 1, we observed a significant upregulation of mir-206 in both denervated and re-innervating muscles after 10 days and 30 days from surgical nerve dissection compared to the control and to 2 days denervated muscles. Conversely, we observed a significant downmodulation of mir-133a after 10 days and 30 days from denervation compared to the control and to 2 days denervated muscles, indicating an opposite trend of expression of the two myomiRs in relation to nervous stimuli.

Skeletal muscle can release tissue-specific microRNAs in the blood stream, and it has been reported that the levels of circulating microRNAs can be modulated in relation to muscle remodeling. Therefore, to evaluate miR-206 and miR-133a release in blood circulation during denervation and re-innervation processes, we performed an absolute quantitative RT-PCR analysis of circulating miR-206 and miR-133a in serum samples of wild type mice 2, 10 and 30 days after nerve dissection, under denervated and re-innervating conditions (Figure 2).

As shown in Figure 2, we observed a significant reduction of both miR-206 and miR-133a after 2 and 10 days from re-innervation, and an increase of miR-206 and miR-133a levels to values comparable to the control mice at 30 days after re-innervation.

Conversely, circulating levels of both miR-206 and miR-133a were comparable to the control mice 2 days after denervation, and were downregulated 10 and 30 days after surgery, compared to the control and to 2 days denervated muscles. Further, comparative analysis of the miR-206 copy number 30 days after surgical nerve dissection revealed higher levels of miR-206 in re-innervating muscle compared to chronic denervated muscle, suggesting that the increase in miR-206 levels is crucial for muscle re-innervation. Differently, we observed that 30 days after surgical denervation, the miR-133a copy number increased independently from nerve stimuli, becoming comparable to the control condition in the re-innervating condition, and remaining significantly lower in the denervated condition.

### 3.2. MiR-206 Is Differently Modulated in Fast and Slow Progressive SOD1^G93A^ Mice

ALS patients display signs of denervation and re-innervation of muscle fibers during disease progression. To evaluate the levels of circulating myomiRs during ALS’ pathological alteration of muscle–nerve communication, we performed an absolute RT-PCR quantification of miR-206 and miR-133a in the serum samples of ALS mouse models at different stages of the disease.

We analyzed the copy numbers of circulating miR-206 at the onset (90 days) and at the end stage (150 days) of the disease in fast progressing SOD1^G93A^ high-copy mice (Figure 3). As shown in Figure 3, we did not observe any difference in miR-206 levels between wild type and SOD1^G93A^ high-copy mice at the onset of the disease, while we observed higher levels of miR-206 at the terminal phase of the disease, compared to both wild type and to SOD1^G93A^ high-copy mice at the onset stage.

Further, to better evaluate whether the velocity of disease progression could be related to the level of miR-206, we took advantage of another transgenic ALS mouse model, namely the SOD1^G93A^ low-copy model, which exhibits a slow progression of the disease. We analyzed the levels of circulating miR-206 at the onset (150 days), at the late (220 days) and at the end stage (250 days) of the pathology (Figure 3); we observed that circulating levels of miR-206 were significantly upregulated at the late symptomatic phase (220 days) compared to wild type control mice and to early symptomatic (150 days) and end stage (250 days) mice, while they were not significantly modulated at 150 and 250 days compared to wild type control mice.

Interestingly, the circulating levels of miR-206 were significantly higher in SOD1^G93A^ high-copy mice compared to age-matched SOD1^G93A^ low-copy mice (150 days), suggesting that different levels of miR-206 can be associated to different velocities of the disease progression. 

Moreover, although miR-206 levels increased during slow progressive disease, miR-206 levels of SOD1^G93A^ low-copy mice were maintained significantly lower throughout the disease, if compared to late symptomatic SOD1^G93A^ high-copy mice.

To evaluate whether the number of copies of miR-206 could discriminate fast from slow progressing ALS mice, we performed a ROC analysis of miRNA levels in SOD1^G93A^ high-copy and low-copy mice (Figure 4a,b). As described in Figure 4a, miR-206 represents a good biomarker for symptomatic ALS mice independent from disease velocity.

### 3.3. MiR-133a Levels Are Positively Modulated during Disease Progression of Both SOD1^G93A^ High-Copy and SOD1^G93A^ Low-Copy Mice

The analysis of miR-133a levels demonstrated a significant increase in the number of miR-133a copies during disease progression in both SOD1^G93A^high-copy and SOD1^G93A^ low-copy mice. In particular, we observed that the serum levels of miR-133a were more highly maintained throughout the disease compared to wild type serum samples, independent from disease velocity (Figure 5). In addition, the number of miR-133a copies was significantly downregulated at the end stage of the disease (250 days) compared to the late stage (220 days) in slow progressive mice, while the number of copies progressively increased during the disease in SOD1^G93A^ high-copy mice.

To evaluate whether the number of copies of miR-133a could discriminate fast from slow progressing ALS mice from control mice, we performed a ROC analysis of miRNA levels in SOD1^G93A^ high-copy and low-copy mice. As described in Figure 6, miR-133a represents a good biomarker for both fast and slow progressive ALS mice at the onset and end stages of SOD1^G93A^ high-copy mice (Figure 6a,b) and at the late and end stages of the disease in SOD1^G93A^ low-copy mice (Figure 6c,d).

### 3.4. Increasing Amounts of miR-206 Are Associated to Different Response Effect

To evaluate whether miR-206 levels were modulated during re-innervation and ALS disease, we compared wild type re-innervating mice and SOD1^G93A^ low-copy and high-copy mice.

The analysis revealed a modulation of circulating miR-206 levels between wild type re-innervating mice and SOD1^G93A^ low-copy and high-copy mice. 

We observed an increasing amount of circulating miR-206 in SOD1^G93A^ low-copy mice (+2.51 fold change vs WT), in wild type re-innervating mice (+7.38 fold change WT REN 30D vs WT REN 2D), and in SOD1^G93A^ high-copy mice (+11.61 fold change vs WT), suggesting that an increase in miR-206 is associated with muscle re-innervation but that different increases in circulating miR-206 are related to the pathological condition and different levels of disease aggressiveness.

Therefore, in order to identify the threshold value of miR-206 that could discriminate the restoration of physiological communication between muscles and nerves from those associated with ALS disease, we performed a ROC curve analysis of the miR-206 levels of wild type re-innervating mice and fast and slow progressive ALS mice. The analysis provided a cut-off <161,000 for REN 30D vs. SOD1^G93A^ low-copy and a cut-off <689,500 REN 30D vs. SOD1^G93A^ high-copy mice (Figure 7a,b). Moreover, the ROC curve analysis revealed that miR-206 levels represent a good biomarker to discriminate fast progressive ALS mice from slow progressive mice (Figure 7**c**).

## 4. Discussion

Loss of muscle and nerve communication is the result of several degenerative conditions, such as nerve injury, aging and neurodegenerative diseases. At the post-transcriptional level, miRNAs are thought to be highly involved in the pathophysiological progress of denervated muscles and circulating levels of myomiRs have been recently described as being modulated during ALS disease progression [5,11]. In this study, we performed a longitudinal comparative analysis of circulating miR-206 and miR-133a in re-innervating and denervated wild type muscles and in ALS fast and slow progressing mice and, for the first time, we measured the absolute copy number of circulating miR-206 and miR133a in all the animal models.

We demonstrated that: (i) circulating miR-206 are sensible to nervous stimuli since their levels increase during re-innervation and decrease in the chronic denervated condition; (ii) circulating miR-133a levels decrease after surgical nerve dissection, independently from nervous stimuli; (iii) miR-206 and miR-133a circulating levels increase during ALS disease, remaining high in fast progressive mice and declining in slow progressive mice; (iv) increasing amounts of miR-206 are associated with different response effects to muscle denervation.

The up-regulation of miR-206 during muscle re-innervation and during ALS disease are in line with previous works [11,13] that describe higher levels of miR-206 in re-innervating muscles and which demonstrate that miR-206 expression can attenuate ALS progression, promoting the regeneration of the neuromuscular junction [6,7,11]. However, we observed higher levels of circulating miR-206 at the end stage of the disease in fast progressing ALS mice, which is when total hind limb paralysis occurs, a result that is in apparent contrast with the protective role of miR-206 on the neuromuscular junction (NMJ), but this can be explained by the differing roles that circulating miR-206 play according to the serum level. Indeed, by absolute quantification, we demonstrated a different modulation of circulating miR-206 levels in ALS mice according to disease velocity: we observed that in fast progressive ALS mice, miR-206 levels rapidly increase during disease progression and remain high until the end of the pathology, while in slow progressive ALS mice, miR-206 levels increase at the beginning of the disease and decline at the late stage of the pathology, which is a finding that is in line with previous observation of the serum of ALS patients [11]. All this evidence indicates that low-copy SOD1^G93A^ mice better recapitulate the progression of human pathology and lead us to suppose that a high-copy number of the SOD1 mutant gene exerts a detrimental effect on disease progression. 

The differences between the results obtained in the ALS mouse model and in ALS patients are emphasized by miR-133a analysis, whose expression in literature exhibits contradictory results [11,14,15]. In this study, we demonstrate the up-regulation of miR-133a serum levels in ALS mice independently from disease velocity; these results are in sharp contrast with our previous analysis of miR-133a in serum samples of ALS patients [11] and with data obtained in denervated and re-innervating wild type mice, both of which exhibit low serum levels of miR-133a throughout the disease, suggesting that circulating levels of miR-133a are strictly correlated to the expression of the SOD1 mutant gene rather than to motor neuron degeneration and ALS disease. 

Another novelty of the present study is the comparative analysis of circulating myomiRs in pathological and surgical denervation. We revealed in all the experimental models the upregulation of miR-206 during re-innervation; interestingly, we demonstrated that higher levels of miR-206—10-fold greater than the control one—are related to higher aggressiveness of ALS pathology, and that in the slow progressive ALS mice, the increase of miR-206 levels was less than 3-fold compared to the control condition. In addition, the re-innervating mice up-regulated the circulating miR-206 by about 7.5-fold compared to the first days of denervation, bringing the miRNAs levels back to the control sham denervated condition 30 days after nerve dissection. In all, these results indicate the hormetic activity of miR-206, suggesting that miR-206 can attenuate or hamper muscle re-innervation according to its different expression in the circulation. Moreover, ROC curve analysis allowed us to identify the threshold level of miR-206 in order to discriminate the protective or detrimental effects of mir-206 during re-innervation; this result can be crucial for exploring new tools to fight ALS disease progression.

## Figures and Tables

**Figure 1 cells-10-02043-f001:**
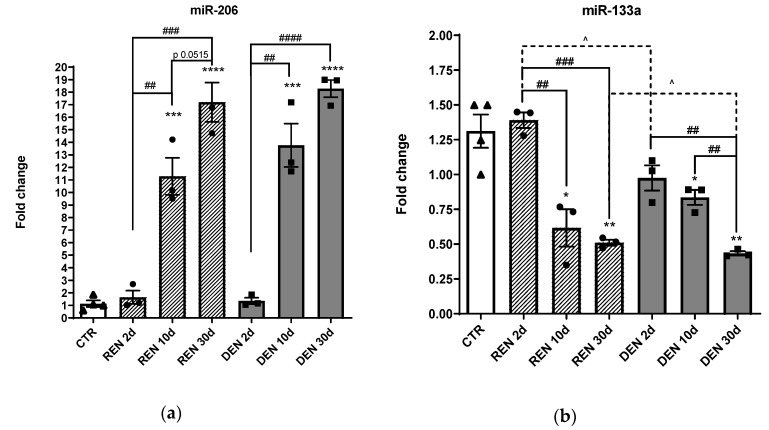
Muscle expression of miR-206 and miR-133a during muscle denervation and re-innervation. RT-PCR of (**a**) miR-206, (**b**) miR-133a in re-innervating (REN) and chronic denervated gastrocnemius muscles (DEN), 2 days (2d), 10 days (10 d) and 30 days (30 d) after sciatic nerve resection (* comparison vs. control: (* *p* < 0.05, ** *p* < 0.005, *** *p* < 0.0005, **** *p* < 0.0001. ^ and # comparison between groups ^ *p* < 0.05, ## *p* < 0.005, ### *p* < 0.0005, #### *p* < 0.0001. •, ▲, ■ represent samples in control, re-innervating and denervated groups).

**Figure 2 cells-10-02043-f002:**
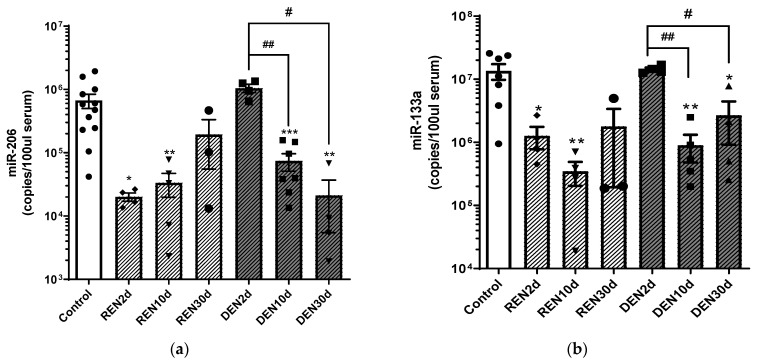
Serum levels of miR-206 and miR-133a during muscle denervation and re-innervation. Absolute quantification of miR-206 (**a**) and miR-133a (**b**) serum levels in re-innervating (REN) and denervated (DEN) gastrocnemius muscles, 2 days (2 d), 10 days (10 d) and 30 days (30 d) after sciatic nerve resection (* comparison vs. control: * *p* < 0.05, ** *p* < 0.005, *** *p* < 0.0005. # comparison between groups: # *p *< 0.05, ## *p*  <  0.005. •, ▲, ▼, ◆, ■, represent samples in control, re-innervating and denervated groups).

**Figure 3 cells-10-02043-f003:**
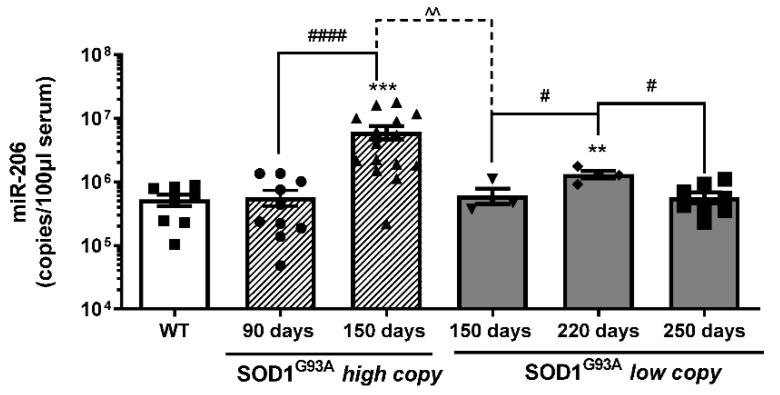
Serum levels of mir-206 are differently modulated during ALS disease progression. Absolute quantification of miR-206 levels in SOD1^G93A^ high-copy and SOD1^G93A^ low-copy mice at different stages of the disease (* comparison vs. control: ** *p* < 0.005, *** *p* < 0.0005. ^ and # comparison between groups: ^^ *p* < 0.005, # *p* < 0.05, #### *p*  <  0.0001. •, ▲, ▼, ◆, ■, represent samples of WT control mice, and SOD1^G93A^ high-copy and SOD1^G93A^ low-copy mice at different stages of the disease).

**Figure 4 cells-10-02043-f004:**
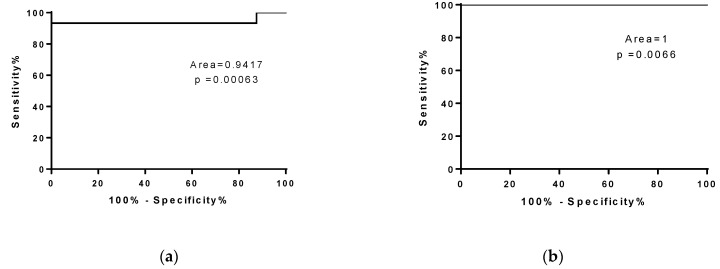
MiR-206 is a biomarker for symptomatic ALS mice. ROC curve analysis of miR-206 levels in WT mice compared to (**a**) SOD1^G93A^ high-copy mice at end stage and (**b**) SOD1^G93A^ low-copy mice at late stage of the disease.

**Figure 5 cells-10-02043-f005:**
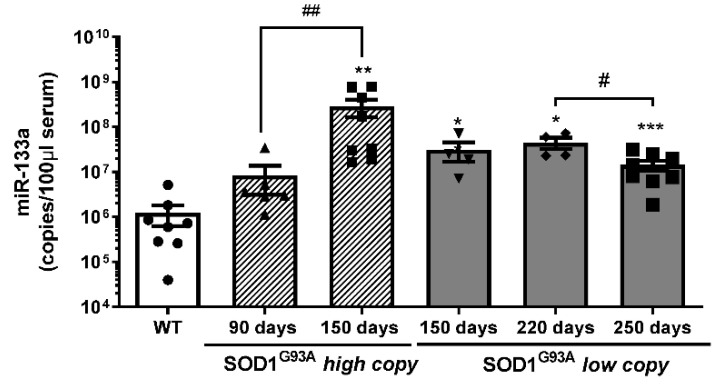
Serum levels of mir-133a are differently modulated during ALS disease progression. Absolute quantification of miR-133a levels in SOD1^G93A^ high-copy and SOD1^G93A^ low-copy mice at different stages of the disease (* comparison vs. control: * *p* < 0.05, ** *p* < 0.005, *** *p* < 0.0005. # comparison between groups: # *p* < 0.05, ## *p* < 0.005. •, ▲, ▼, ◆, ■, represent samples of WT control mice, and SOD1^G93A^ high-copy and SOD1^G93A^ low-copy mice at different stages of the disease).

**Figure 6 cells-10-02043-f006:**
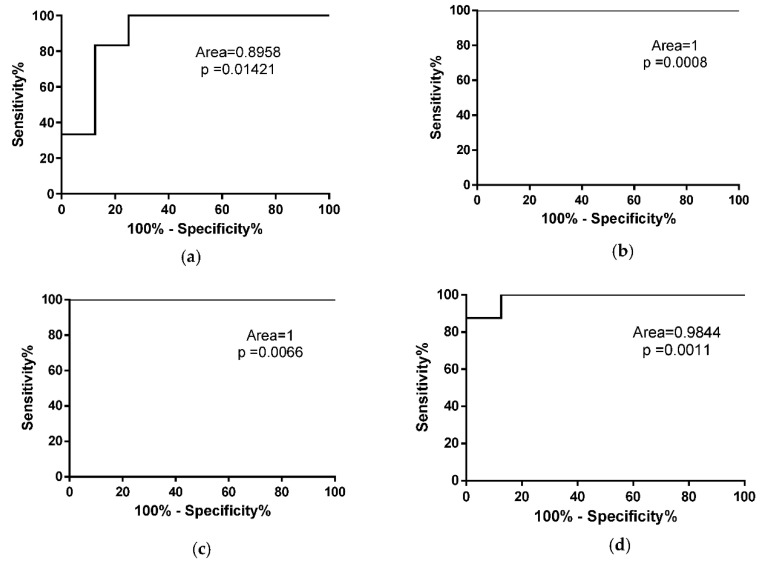
miR-133a is a biomarker for symptomatic ALS mice. ROC curve analysis of miR-133a levels in WT mice compared to SOD1^G93A^ high-copy mice at (**a**) early stage and (**b**) end stage of the pathology. (**c**,**d**) ROC curve analysis of miR-133a in WT mice compared to SOD1^G93A^ low-copy mice at late stage (**c**) and end stage (**d**) of the disease.

**Figure 7 cells-10-02043-f007:**
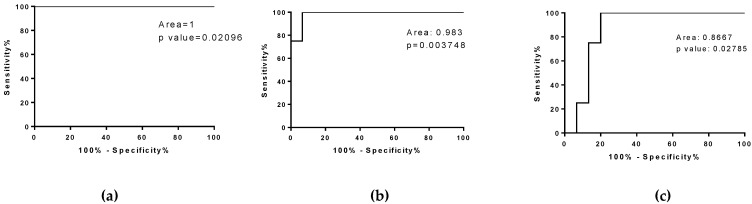
ROC curve analysis of mir-206 in fast and slow progressive SOD1^G93A^ mice and in WT re-innervating mice. (**a**,**b**) ROC curve analysis of miR-206 level in WT REN 30D compared to (**a**) SOD1^G93A^ low-copy mice at the late stage of the pathology and compared to (**b**) SOD1^G93A^ high-copy mice at end stage of the pathology. (**c**) ROC curve analysis of miR-206 level in SOD1^G93A^ high-copy mice compared to SOD1^G93A^ low-copy mice, respectively, at the end stage and late stage of the pathology.

## Data Availability

The datasets used to support the findings of this study are included in the present article and are available from the corresponding author upon request.

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
