# Peer review of "Circulating myomiRs in Muscle Denervation: From Surgical to ALS Pathological Condition"

_cells, 2021, doi:10.3390/cells10082043_

Round 1

Reviewer 1 Report

The authors analyse the role of microRNA ( miR-206 and miR-133a ) in denervation / re-innervation processes and in a mouse ALS models. ( SOD1 G93A )

The manuscript is clear and well written and the topic certainly interesting to a wide readership. The experimental strategy is strait forward ,  the data and its  analyses (  eg statistics ) are convincing.  The presented findings are however not totally novel ; the role of ( the same ) micro-RNAs during denervation have already been studied in other model  and the proposed use of them as biomarkers for ALS have already been studied in patients ( also by the authors ) 

The use here of two ALS models  ( high and low copy SOD 93 ) is however a real plus of our knowledge on ALS and miRNA .

I have further a problem with the used terms . Under the reported experimental conditions you cannot observe re-innervation , at least not in the reported time-frame . Sprouting eventually  but certainly no re-innervation of neuro-muscular junctions ( NMJ ).  What happens should be documented by histologically or biochemically ( PCR of ACh receptor subunits) analyses of NMJ or documented by functional studies ( eg  recovery of muscle strength )

Whatever the final decision , this point must be discussed and presented more clearly . 

Reviewer 2 Report

The manuscript by Casola and colleagues explores the role of miR-206 and miR-133a in ALS and give new insights in their role as myomiRs participating in the homeostasis of muscle re-innervation process.

The study is well designed and the conclusions drawn from the paper are important and novel in this field.

In my point of view, there are still some major points that should be addressed in order to improve the quality of the manuscript, as it follows:

- The manuscript should be checked in order to improve English, namely the verbs in the results section and in the end of the introduction, that should be on in past tense and not in the present.

- It is not clear whether miR-133a is up or downregulated depending on the ALS model used, type of sample or disease progression stage. In fact, contradictory reports have emerged in this regard. Such information deserves to be further discussed in the manuscript.

- The Authors use the SOD1-G93A mice, either with high copy and low copies of the mutation, In my point of view, it should be more detailed how the Authors have confirmed that this feature correlates with the speed of disease progression, as claimed in the manuscript. Some information about the behavior and/or molecular profile on the two models used will be important to understand the different effects here presented in terms of the miRNAs studied.

Reviewer 3 Report

1. The mRNA indicators related to ALS are not only 206 and 133a, such as MIR206, MIR208b and MIR499 are also common indicators, so why specifically choice two of these miRNAs? 2. The article mainly provides miRNAs in animal muscle tissue and serum, but the trends of the two are not consistent. How to explain this phenomenon? 3. Please write clearly about statistics in methodology 4. Please explain the main reason why miRNA-133a is significantly decrease before RED30 or DEN30 and increased in the blood regardless of REN30 or DEN30 5. Please explain why it is shown in Figure 5 that the expression of miR-133a in SOD1 low-copy mouse traces does not decrease until 250 days later. What does it mean? 6. The article stated that the SOD1 gene has an effect on miRNA-133a. In the past, there were also articles describing that miR-133a cells have a significant effect and affect the life cycle of mice. In animal experiments, no specific differences in the life cycle length of life have been observed. 7. Why not choose to collect data directly on patients with ALS? How should these findings be applied to ALS disease prediction in the future?

Round 2

Reviewer 2 Report

I have no further questions and recommend acceptance.

Reviewer 3 Report

This is an article with clinical significance. Thank you for your careful modification, please modify it carefully, and smooth the sentence and picture description, so that readers can understand your article better